# Examining Obedience Training as a Physical Activity Intervention for Dog Owners: Findings from the Stealth Pet Obedience Training (SPOT) Pilot Study

**DOI:** 10.3390/ijerph18030902

**Published:** 2021-01-21

**Authors:** Katie Potter, Brittany Masteller, Laura B. Balzer

**Affiliations:** 1Department of Kinesiology, University of Massachusetts Amherst, Amherst, MA 01003, USA; 2Department of Exercise & Sport Studies, Smith College, Northampton, MA 01063, USA; bmasteller@smith.edu; 3Department of Biostatistics and Epidemiology, University of Massachusetts Amherst, Amherst, MA 01003, USA; lbalzer@umass.edu

**Keywords:** dog walking, exercise, health behavior change, stealth health, pet ownership, human–animal interaction, animal-assisted intervention, targeted learning

## Abstract

Dog training may strengthen the dog–owner bond, a consistent predictor of dog walking behavior. The Stealth Pet Obedience Training (SPOT) study piloted dog training as a stealth physical activity (PA) intervention. In this study, 41 dog owners who reported dog walking ≤3 days/week were randomized to a six-week basic obedience training class or waitlist control. Participants wore accelerometers and logged dog walking at baseline, 6- and 12-weeks. Changes in PA and dog walking were compared between arms with targeted maximum likelihood estimation. At baseline, participants (39 ± 12 years; females = 85%) walked their dog 1.9 days/week and took 5838 steps/day, on average. At week 6, intervention participants walked their dog 0.7 more days/week and took 480 more steps/day, on average, than at baseline, while control participants walked their dog, on average, 0.6 fewer days/week and took 300 fewer steps/day (difference between arms: 1.3 dog walking days/week; 95% CI = 0.2, 2.5; 780 steps/day, 95% CI = −746, 2307). Changes from baseline were similar at week 12 (difference between arms: 1.7 dog walking days/week; 95% CI = 0.6, 2.9; 1084 steps/day, 95% CI = −203, 2370). Given high rates of dog ownership and low rates of dog walking in the United States, this novel PA promotion strategy warrants further investigation.

## 1. Introduction

Fewer than one in four American adults meet federal physical activity (PA) guidelines [1]. Part of the problem may be messaging around PA. Traditional messaging frames PA as a critically important health behavior that people should engage in to improve health and prevent disease. While this may motivate PA adoption short-term, this framing provides a controlled source of motivation for PA (in the form of external pressure to be healthy), and therefore is unlikely to motivate consistent participation over time [2,3]. Sustainable PA might be more realistically achieved by encouraging people to engage in activities they enjoy, care about, or find purposeful and that naturally involve PA (i.e., that increase PA as a side-effect) [4,5]. This approach preserves autonomy and therefore better aligns with the science of motivation and decision-making [2]. Interventions that take this approach are called “stealth interventions” [4,5]. Stealth interventions have been successful in promoting healthy eating and weight control [6,7,8].

A relevant aspect of American culture that lends itself well to stealth PA interventions is dog ownership. Almost half of households in the United States (46%) own at least one dog [9]. Research has demonstrated that owners who walk their dog(s) are more likely to meet PA guidelines than those who do not [10]. The strength of the dog–owner bond is a key correlate of dog walking behavior [11,12], as owners who have a strong relationship with their dog(s) feel a greater sense of responsibility to walk and perceive more motivation and support from their dog(s) for walking [11,13]. A recent national survey study found that only 42% of American dog owners walk their dog [14], suggesting a potential target for stealth PA interventions.

Interventions designed to strengthen the dog–owner bond are already offered in communities across the United States. These interventions are obedience training classes. In addition to strengthening the dog–owner bond [15,16], these classes teach basic manners, including loose leash walking, which may increase dog-walking self-efficacy [17] and reduce behavior problems that can interfere with dog walking [13]. To the best of our knowledge, dog obedience training classes have never been examined in the context of PA promotion.

The purpose of the Stealth Pet Obedience Training (SPOT) randomized trial was to pilot a six-week, basic obedience training class as a stealth PA intervention for inactive dog owners (NCT04329741). Again, the logic behind this approach is that some dog owners who take the course may naturally develop a new, personal source of motivation for PA (in the form of dog walking) as a result of becoming more attached to their dog through the dog training experience. We consider this a stealth approach not because dog owners will be unaware that they are engaging in more PA, but because the intervention is a course focused on improving dog obedience, not increasing PA. We hypothesized that the intervention would lead to greater increases in average steps/day and daily minutes of moderate–vigorous PA (MVPA) at 6 and 12 weeks, as compared to the waitlist control. Secondary outcomes included dog walking days per week, daily sedentary minutes, and psychosocial variables plausibly mediating PA changes: strength of dog–owner bond (emotional closeness with dog), dog walking self-efficacy, and social support (from the dog) for walking.

## 2. Materials and Methods

### 2.1. Study Design and Population

The SPOT study was an individually randomized trial with a waitlist control group. Adult dog owners (21+ years) who reported walking their dog(s) ≤3 times/week (for no more than 20 min/walk) were eligible for participation. Exclusion criteria were regular exercise (defined as ≥3 times per week for ≥20 min), previously attending obedience training with their current dog, presence of any condition that limits walking ability, or presence of uncontrolled diabetes or hypertension. These criteria were set to identify a sample of inactive dog owners who could safely walk for exercise. All participants provided informed consent. This study was approved by the Institutional Review Board (protocol ID: 2017-3945) and the Institutional Animal Care and Use Committee (protocol ID: 2017-0018) at the University of Massachusetts-Amherst. Randomization was conducted by an external researcher.

### 2.2. Procedures

Participants were recruited through university-affiliated social media outlets. The study was advertised as an investigation of how attending a dog obedience training course affects the dog–owner bond and the health and quality of life of dog owners. Participants randomized to the intervention group enrolled in a six-week basic obedience training class led by a certified behavior adjustment trainer. The class covered basic commands (e.g., sit, down, watch), loose leash walking, and polite greetings, among other skills. The importance of dog walking was implied, but not specifically emphasized. Classes were held once per week for 45 min, with 5–8 students per class. Participants in the control group were asked not to enroll in an obedience training class or train their dog at home until after completing 12-week assessments. After completing these assessments, control group participants received a voucher to take the same obedience training class for free. Baseline measures were taken in August, six-week measures in October, and 12-week measures in December 2017 in Massachusetts, United States.

### 2.3. Measures

Participants reported their demographics (e.g., age, sex), characteristics of their dog (e.g., age, size), and whether they had a yard (“Do you have a yard where your dog can run free?” (yes/no)). To estimate dog size, participants were given four options (small, medium, large, giant) and provided example dog breeds for each option. Research staff measured each participant’s height and weight at orientation for body mass index calculation.

#### 2.3.1. Process Evaluation

The study process was evaluated through the following metrics. Retention was assessed as the percentage of randomized participants who completed 12-week assessments. Class attendance, defined as the average number of classes attended across the 6 weeks, served as the indicator of intervention engagement. Intervention fidelity was indicated by the proportion of intervention participants who agreed or strongly agreed with the prompt “I am happy with the behavior of my dog” at week 6 compared to control participants.

#### 2.3.2. Physical Activity and Sedentary Behavior

For seven consecutive days at baseline, post-program (at 6 weeks), and 6 weeks post-program (at 12 weeks), participants wore an ActiGraph wGT3X-BT monitor (ActiGraph LLC, Pensacola, FL, USA) on their right hip [18,19,20] and logged all leisure-time PA, including dog walking, in a paper log booklet. The ActiGraph, a research-grade triaxial accelerometer deemed valid [21] and reliable [22] in free-living conditions, was used to assess changes in steps, MVPA, and sedentary behavior. Participants wore the device during all waking hours, except when showering/swimming.

ActiGraph data were processed using Actilife Version 6.13.3 (ActiGraph LLC, Pensacola, FL, USA); validated cut-points for adults [23] were used to evaluate minutes spent in different PA intensity categories. To be included in analyses, participants had to wear the device ≥8 h/day for ≥4 days, including one weekend day; otherwise, data were considered missing [24]. Participant data were averaged across valid wear-days to produce daily estimates.

#### 2.3.3. Psychosocial Outcomes

The 10-item perceived emotional closeness subscale from the Cat/Dog–Owner Relationship Scale (C/DORS) was used to assess the dog–owner bond [25]. Each item is scored on a five-point scale (from 1–5). Sample items include “My pet provides me with constant companionship” and “My pet is there whenever I need to be comforted”. The 10 item scores were summed and divided by 10 to yield a subscale score ranging from 1–5, with higher scores indicating better perceived relationship quality.

Subscales from the Dogs and Walking Survey (DAWGS) [26] were used to assess self-efficacy beliefs about dog walking and support provided by the dog for walking. The self-efficacy subscale consists of nine items from the Exercise Confidence Survey [27] modified for dog walking. Participants are asked to rate how confident they are that they would consistently walk their dog under a number of circumstances if they really wanted to (1 = very unconfident to 5 = very confident). Scores on the nine items were summed to produce a dog walking self-efficacy score ranging from 9–45. Three five-point items (1 = strongly disagree to 5 = strongly agree) were used to assess dog support for walking. These items were “Having my dog makes me walk more”, “My dog provides encouragement for me to go on walk”’, and “My dog provides social support for me to go on walks”. Scores were summed to produce a total dog support for walking score ranging from 3–15.

### 2.4. Statistical Analyses

A sample size of 40 adults was selected to provide at least 80% power to detect differences between randomized groups of at least 1500 average steps/day and at least 76 min/week of MVPA. Such differences are consistent with meeting 50% of aerobic physical activity guidelines, which have been associated with improvements in cardiorespiratory fitness among previously sedentary individuals [28,29].

We compared 6- and 12-week outcomes between randomized arms using targeted maximum likelihood estimation (TMLE), which provides precision and power gains over an unadjusted approach (e.g., the Student’s *t*-test) in randomized trials [30,31]. We used a pre-specified adjustment strategy and excluded participants whose outcome assessments were missing at the timepoint of interest [32]. All PA outcomes were parameterized in terms of the change from baseline. Statistical inference was based on the *t*-distribution and with a two-sided hypothesis test at the 5% significance level. All analyses were completed with R v3.5.1. 9. (The R Foundation, Vienna, Austria). Additional details about the statistical approach are provided in the Appendix A.

## 3. Results

The median age of participants was 37 years, and most were non-Hispanic White (87%) and female (87%) (Table 1). The median age of the study dogs was 3 years. About half the dogs were large (50–90 lbs; *n* = 18), followed by medium-sized (20–49 lbs; *n* = 10), small/toy-sized (<20 lbs; *n* = 7), and giant-sized (>90 lbs; *n* = 4). At baseline, participants averaged 5838 steps/day, 22 MVPA minutes/day, and reported dog walking 1.9 days/week (Table 1).

Four of the 21 participants randomized to intervention dropped out before or during the six-week class. All of the remaining 17 intervention participants completed the 6-week assessments, and 16 of 17 completed 12-week assessments. All 20 control participants completed both 6- and 12-week assessments. Altogether, 36 of 41 randomized participants completed 12-week assessments for an overall study retention rate of 88%. Intervention participants attended an average 5.6 out of 6 classes and were 33% more likely to agree or strongly agree with the prompt “I am happy with the behavior of my dog” than control participants at week 6 (95% CI = 6%,60%).

ActiGraph wear time criteria were met by 38 participants (19 per group) at baseline, 34 participants (16 intervention, 18 control) at 6 weeks, and 33 participants (16 intervention, 17 control) at 12 weeks. At 6 weeks, intervention participants took 480 more steps/day than they had at baseline, while control participants took 300 fewer steps/day than at baseline (Figure 1). Thus, the difference in the average change in steps/day between randomized arms was 780 steps/day (95% CI = −746, 2307). These differences persisted at 12 weeks, when intervention participants had essentially no change in their steps/day from baseline, and control participants decreased by 1084 steps/day for an average difference of 1084 steps/day (95% CI = −203, 2370).

These changes in daily steps were echoed in changes in daily MVPA minutes (Figure 1). At 6 weeks, intervention participants averaged 4.7 more MVPA minutes than they had at baseline, while control participants averaged <1 more MVPA minute than at baseline, for a difference of 4.3 min (95% CI = −7.8, 16.4). At 12 weeks, both intervention and control participants had decreased their MVPA minutes by 1.9 and 3.1 min from baseline, respectively (difference = 1.2; 95% CI = −6.4, 8.8).

Self-reported days with at least one dog walk are reported in Table 2. At 6 weeks, intervention participants walked their dog, on average, 0.7 more days/week than they had at baseline, while control participants walked their dog, on average, 0.6 fewer days/week (difference = 1.3; 95% CI = 0.2, 2.5). At 12 weeks, intervention participants increased their dog walking by 0.9 more days/week compared to baseline, and control participants reduced their dog walking by 0.8 days/week (difference = 1.7; 95% CI = 0.6, 2.9).

Trends were also observed in daily sedentary minutes (Table 2). At 6 weeks, intervention participants had increased their sedentary time by 2.1 min on average, while control participants had increased their sedentary time by 16 min on average (difference = −13.9; 95% CI = −41.2, 13.4). By 12 weeks, intervention participants had increased their daily sedentary time by <1 min from baseline, whereas control participants increased their sedentary time by a half-hour (30.2 min), for an average difference of −29.3 min/day (95% CI = −69.8, 11.2).

Intervention effects on psychosocial outcomes are shown in Table 2. At 6 and 12 weeks, both groups reported similar levels of emotional closeness with their dog. Intervention participants averaged 3.8–3.9 out of 5 possible points on the C/DORS emotional closeness subscale, whereas control participants averaged 4.0–4.1 points. Both groups also reported similar social support from the dog for walking; at 6 weeks, intervention participants averaged 11.2 out of a maximum 15 points compared to an average 10.4 points in the control arm. Intervention participants did, however, report higher confidence in their ability to walk the dog in the face of barriers; at 6 weeks, their average score was 31.9 (maximum score of 45), compared to an average score of 29.0 in the control group (difference = 2.9; 95% CI = −0.7, 6.4). Similar trends were observed at 12 weeks (difference = 7.4; 95% CI = −1.2, 15.9).

## 4. Discussion

The SPOT study piloted a stealth health approach to increasing PA among inactive dog owners. Immediately post-intervention (at 6 weeks), the intervention group averaged more steps and MVPA minutes per day than the waitlist control group, although differences were modest (780 steps/day and 4.7 MVPA minutes/day, on average). The modest group difference in daily steps was maintained at the follow-up assessment (1080 steps/day, on average), and was driven by a decrease in steps by control participants. Given the timeline of the study (i.e., baseline measures taken in summer, 12-week measures taken in winter), a plausible explanation for these findings is that obedience training buffered against a decrease in leisure-time PA with the onset of winter [33], which was observed in the control group.

Most existing interventions to increase dog walking focus on health benefits for the dog and/or owner [34,35,36,37] and promote dog walking as something owners should do. In contrast, the approach tested in this study aimed to strengthen the dog–owner bond and foster a sustainable PA habit as a side effect (small sample sizes prohibited formal mediation analyses). Importantly, basic dog obedience training is already available in communities across the United States, and to ensure the potential for population-level dissemination, the program was not modified in any way. The success of this pilot was reflected in the high rate of class attendance and low study attrition. Furthermore, intervention participants were more likely to be happy with their dogs’ behavior at 6 weeks compared to control participants, demonstrating that the course was effective in achieving its primary purpose.

Increases in steps in the range of 1000–2000 steps/day have been associated with reduced risk of type II diabetes [38], cardiovascular disease [39,40], and all-cause mortality [41,42,43]. Immediately post-intervention, there were average daily increases of nearly 500 steps and 5 MVPA minutes (the equivalent of 500 steps taken at a rate of ≥100 steps/min [44]) among intervention participants. Notably, nearly a third (31%) of intervention participants increased their steps/day by at least 1000 steps at both 6 and 12 weeks. In comparison, PA interventions that use pedometers to increase walking behavior have led to short-term increases of about 2000 steps/day [45]. Altogether, the modest changes in PA shown in the SPOT pilot are promising given that the intervention was not advertised for PA and did not specifically focus on dog walking. This stealth approach may reach inactive adults who are uninterested in or unmotivated by traditional PA promotion efforts, and therefore modest intervention efficacy may translate to large public health impact [46].

Intervention participants also accumulated fewer daily sedentary minutes than control participants immediately post-intervention and at the 12-week follow-up assessment. The group differences in sedentary behavior were driven by decreases from baseline among control participants, which may have been due to seasonal changes. Since intervention participants were asked to work on new skills outside of class, it is plausible that more time spent training or playing with one’s dog (irrespective of walking) buffered against an increase in sedentary behavior with the onset of winter. Although the best sedentary behavior interventions to date primarily aim to change sedentary behavior (not PA) [47,48], this finding suggests that obedience training could positively impact both PA and sedentary behavior among dog owners. A few studies have examined dog ownership in relation to sedentary behavior [49,50,51], but no intervention studies have attempted to leverage the dog–owner bond to reduce sedentary behavior.

We anticipated that changes in PA would be driven by differences in the dog–owner bond, but did not observe meaningful differences in the dog–owner relationship. All participants were highly attached to their dog at baseline, and therefore a ceiling effect may have occurred. It is also possible that intervention participants did experience increased feelings of attachment toward their dog, but that the sample size was too small or the questionnaire not nuanced enough to detect changes. Qualitative data collection may be helpful for measuring changes in the human–animal bond in this study population.

Intervention participants did, however, report higher dog walking self-efficacy than control participants at 6 and 12 weeks. The self-efficacy construct is actively debated, with some experts suggesting it is confounded with motivation [52,53]. In SPOT, participants were asked to rate their confidence in their ability to consistently walk their dog under a number of circumstances (e.g., after a long day at work, when undergoing a stressful life change) if they really wanted to. The “if you really wanted to” qualifier attempts to assess perceived capability independent of motivational factors [54]. We attribute higher self-efficacy for dog walking among intervention participants to the mastery of program skills, including loose leash walking and the “leave it” command. However, it is also possible that the acquisition of these skills made dog walking more enjoyable, and this greater enjoyment led to greater motivation to walk.

The major strength of this study is the innovative stealth health approach tested. Stealth approaches may engage individuals who are uninterested in lifestyle change for better health and disease prevention (and therefore would not participate in traditional interventions), and may promote sustainable change given that the change is personally meaningful or enjoyable. The stealth approach tested in this study has potential for high reach as, by some estimates, nearly 50% of American households have a dog [9]. Furthermore, dog walking may be a particularly sustainable form of PA to promote, as it serves a purpose and lends itself well to habit formation [55]. Other strengths of this study include the rigorous randomized design and use of objective measures of PA.

Limitations of this study include its small sample size, which led to large variability in PA outcomes. Baseline assessments were held in summer, and follow-up assessments were held in winter; therefore, the study timeline corresponded with major changes in weather and lifestyle (i.e., back to school) that may affect PA levels. The small sample size in concert with these seasonal changes likely made intervention-driven changes in PA difficult to detect. Finally, although the researchers aimed to determine whether changes in PA naturally occurred as a side-effect of attending a dog training course, participants were aware that the research team was interested in changes in PA and health. This transparency was required for both ethical and practical reasons (i.e., participants wore an accelerometer to provide PA data), but may have led to an expectancy effect. Therefore, changes in PA may not have been driven entirely by autonomous motivation developed as a side-effect of dog training, as theorized.

## 5. Conclusions

The results of this pilot, randomized trial suggest that attending a basic dog obedience training course may lead dog owners to walk more and sit less. If the positive effects observed in this trial are replicated in larger trials with longer follow-up periods, interdisciplinary partnerships can work to normalize and incentive obedience training among new and current dog owners, and to increase accessibility for low-income owners. Given the high rate of dog ownership [9] and low rate of dog walking in the United States [14], this novel approach to PA promotion has the potential for considerable public health impact and warrants further investigation.

## Figures and Tables

**Figure 1 ijerph-18-00902-f001:**
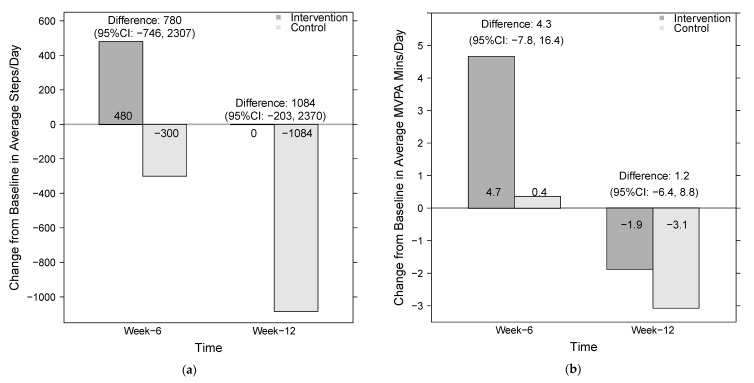
Change in average steps/day (**a**) and moderate-to-vigorous physical activity (MVPA) minutes/day (**b**) among participants in the Stealth Pet Obedience Training (SPOT) pilot study from baseline to 6 and 12 weeks. Analysis restricted to participants with valid ActiGraph data at both baseline and timepoint of interest: 16 intervention participants and 18 control participants at week 6, and 16 intervention participants and 17 control participants at week 12.

**Table 1 ijerph-18-00902-t001:** Baseline characteristics of Stealth Pet Obedience Training (SPOT) study participants and their dogs.^1^

	Overall (*n* = 39)	Intervention (*n* = 19)	Control (*n* = 20)
Age, median (min–max) in years	37 (21–72)	39 (27–72)	34 (21–54)
Sex, *n* (%) female	34 (87%)	16 (84%)	18 (90%)
Race, *n* (%) non-Hispanic White	34 (87%)	18 (95%)	16 (80%)
Annual income, *n* (%)			
<$40,000	5 (13%)	4 (21%)	1 (5%)
$40,000—$80,000	16 (41%)	5 (26%)	11 (55%)
>$80,000	18 (46%)	10 (53%)	8 (40%)
Education, *n* (%)			
High school or GED	8 (21%)	3 (16%)	5 (25%)
College degree	15 (38%)	8 (42%)	7 (35%)
Graduate or professional degree	16 (41%)	8 (42%)	8 (40%)
Body mass index, median (min–max) in kg/m^2^	30 (20.5–44.8)	30 (20.5–44.8)	29.8 (22.6–42.3)
Dog’s age, median (min–max) in years	3 (0–11)	4 (0–10)	2.8 (0–11)
Dog’s size, *n* (%)			
Giant (>90 lbs)	4 (10%)	2 (11%)	2 (10%)
Large (50–90 lbs	18 (46%)	11 (58%)	7 (35%)
Medium (20–49 lbs)	10 (26%)	5 (26%)	5 (25%)
Small/toy (<20 lbs)	7 (18%)	1 (5%)	6 (30%)
Yard where dog can run freely, *n* (%)	25 (64%)	15 (79%)	10 (50%)
Agree or strongly agree with prompt “I am happy with the behavior of my dog”, *n* (%)	8 (21%)	3 (16%)	5 (25%)
Days/week with at least 1 dog walk, mean (SD)	1.9 (2.1)	1.7 (1.9)	2.2 (2.4)
Steps/day, mean (SD)	5838 (2141)	5840 (2132)	5836 (2208)
Moderate-to-vigorous physical activity (MVPA) minutes/day, mean (SD)	22 (14)	22 (16)	21 (13)
Sedentary minutes/day, mean (SD)	542 (87)	544 (68)	540 (104)
Emotional closeness with dog, median (min–max) ^2^	3.9 (2.4–5)	3.7 (2.5–4.9)	4.2 (2.4–5)
Social support from dog for walking, median (min–max) ^3^	11 (3–15)	11 (3–15)	11 (7–15)
Self-efficacy for dog walking, median (min–max) ^3^	29 (9–45)	29 (9–45)	30 (18–45)

^1^ Excludes two intervention participants who dropped out prior to study start. Missing ActiGraph data (steps, MVPA, sedentary minutes) for one participant. ^2^ Emotional closeness score from cat/dog–owner relationship scale (C/DORS)—scale 1–5. ^3^ Social support from dog for walking (scale 3–15) and self-efficacy for dog walking (scale 9–45) from dogs and walking scale (DAWGS).

**Table 2 ijerph-18-00902-t002:** Intervention effects on secondary outcomes in the Stealth Pet Obedience Training (SPOT) pilot study.

	At Week 6			At Week 12		
	Intervention Average	Control Average	Difference (95% CI)	Intervention Average	Control Average	Difference (95% CI)
Change in dog walking days/week ^1^	0.73	−0.62	1.35 (0.21, 2.49)	0.92	−0.83	1.75 (0.58, 2.91)
Change in sedentary minutes/day ^1^	2.10	16.00	−13.90 (−41.20, 13.40)	0.90	30.20	−29.30 (−69.80, 11.20)
Emotional closeness with dog ^2^	3.90	4.00	0 (−0.30, 0.20)	3.80	4.10	−0.30 (−0.50, 0)
Social support from the dog for walking ^3^	11.20	10.40	0.70 (−0.90, 2.30)			
Self-efficacy for dog walking ^3^	31.90	29.00	2.90 (−0.70, 6.40)	23.30	15.90	7.40 (−1.20, 15.90)

^1^ Changes at both week 6 and week 12 are compared to baseline. ^2^ Emotional closeness from cat/dog owner–relationship scale (C/DORS); scores can range from 1–5. ^3^ Social support from dog for walking (score range 3–15) and self-efficacy for dog walking (score range 9–45) from dogs and walking scale (DAWGS). Social support for dog walking data not available at 12 weeks.

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
