# Peer review of "Examining Obedience Training as a Physical Activity Intervention for Dog Owners: Findings from the Stealth Pet Obedience Training (SPOT) Pilot Study"

_ijerph, 2021, doi:10.3390/ijerph18030902_

Round 1

Reviewer 1 Report

Please see file attached

Reviewer 2 Report

This is an interesting study on how to use dow ownership tp increase PA. Since the authors write that this is only a pilot I do wonder if you have a larger planned ?

Also, do you have any suggestion on why there was only a small change in steps and why it was a recession among controls. 

There is also the question on demand, i.e. if there are SPOT trainings to be held for all dog owners. 

Another thing is that these kind of interventions seems to qualify as a some kind of nudge. Is this something you have discussed and do you see ethical challenges to offer these stealth interventions where study subject not are informed of the purpose of the intervention? 

Also, what year was the intervention undertaken?

Otherwise a nice and interesting manuscript. 

Reviewer 3 Report

INTRODUCTION

  • The manuscript is not currently persuasive enough for me to be certain that there is a significant contribution. Lack of conceptual details and clear rationale (e.g., the association between motivation and dog walking physical activity).
  • It is unclear what the real objective is: increasing physical activity among dog owners our increase dog owner – dog bound? (line 57 vs line 83). Please clarify.
  • Line 51-52: Bold statement. Provide research that supports this assumption.
  • Line 62: How can dog owners engage in autonomous physical activity when they are proposed to be part of a trial, which is considered as an external source of motivation?

METHODS

  • Provide a rationale for excluding individuals with diabetes and hypertension. As it stands, it is unclear why these individuals could not participate in the study.
  • Did the authors conduct a priori to work out your desired sample size? If no, why not?
  • Please state your sampling method for transparency.
  • Has this study been registered? Provide trial number.
  • Why de BMI calculation? There is no rationale in the introduction section for this measure
  • I question the use of a t-test for statistical analysis. Considering the two-arm trial with 3-time points, why not examine the first normal distribution and then test for repeated measure tests?
  • In addition, the effect size should be calculated.

RESULTS

  • Please report a normal distribution test at baseline for transparency.
  • Report motives for drop-out or chart flow for clarity.
  • Line 157: The authors report 41 participants. But in table 1, there are 39 participants in total. Please clarify.
  • Table 2: not all CI are significant as they include zero. Thus, some of the statements are not statistically representative of significance as stated by the authors. As it stands, on change in dog walking days/week at weeks 6 and 12 displayed significant differences between IG and CG.

DISCUSSION

  • I will not comment much more on the Discussion as I believe it will change considerably after carefully reviewing my comments (i.e., statistical tests, results).
  • Overall, it needs to be more about the implications and interpretation of the findings rather than reintegrating what was described in the introduction and results section.
  • In addition, it should be more theory-driven considering previous works on PA with dog walking.

CONCLUSION

  • After reading your study, I am questioning: so, what? What implications does this study have for practitioners or scholars? As it stands, it seems relatively clear that obedience classes would increase PA. In addition, it would be interesting to compare results with the control group at follow-up, since they received a voucher for free obedience classes. Thus, in comparison with the IG group during the research protocol is paramount.

Reviewer 4 Report

Dear Authors,

Please find attached a summary of my suggestion/revisions of the present manuscript.

In my opinion, only a few minor revisions are needed.

I would suggest improving the scientific soundness of the manuscript avoiding colloquialisms and sentences such as (line 51) "They cost about as much as a FitBit, and much less than an annual gym membership". Please bear in mind that the readers might be from everywhere and an annual membership could considerably vary its cost from place to place, even considering the USA only, and this specific phrasing does not provide any supportive data for the reader's comprehension.

Lines 60-63 describe the effect of the proposed stealth approach as more referred to personal reasons than better health itself, which could be a valid explanation. However, the study has been advertised as an investigation of how strengthening the dog-owner bond affect health (lines 82-83), which contradicts the aforementioned sentence. Please rephrase line 60-63 in order to be consistent.

Please consider adding a few details about the targeted maximum likelihood estimation technique in the statistical analyzes paragraph. I have noticed the interesting reference you cited [25] at this concern, however, the reader’s attention may require further support throughout the main text.

Results are clearly reported and the discussion section frankly analyzes your findings, considering both strengths and limitations. I would suggest adding the comparison term for the 12-week time point in table 2, lines 180-181 and also in the abstract. Are you referring on the baseline or 6 weeks while analyzing data at 12 weeks? I guess all findings are expressed in terms of change from baseline (as you have reported in the statistical analyzes paragraph) but it might be worthy clarify it when reporting specific results as you have done for the 6-week time point.

Title and abstract fairly give an insight into the focus of the current manuscript, highlight the major points.

Best regards

Round 2

Reviewer 3 Report

I would like to amend the authors for their effort in reviewing their research.

Thank you for clarifying some statements.